# DNA Barcoding of the Genus *Magnisudis* (Aulopiformes: Paralepididae) with a Coastal Record and Biological Features of *Magnisudis atlantica*

**DOI:** 10.3390/biology12030349

**Published:** 2023-02-22

**Authors:** Rafael Bañón, Bruno Almón, Sonia Rábade, María Berta Ríos, Alejandro de Carlos

**Affiliations:** 1Servizo de Planificación, Consellería do Mar, Xunta de Galicia, Rúa dos Irmandiños s/n, 15701 Santiago de Compostela, Spain; 2Grupo de Estudo do Medio Mariño (GEMM), Edif. Club Naútico Bajo, 15960 Ribeira, Spain; 3Centro Oceanográfico de Vigo (IEO, CSIC), Subida a Radio Faro, 50-52, 36390 Vigo, Spain; 4Instituto de Investigaciones Marinas (IIM-CSIC), Rúa Eduardo Cabello 6, 36208 Vigo, Spain; 5Cofradía de Pescadores San José de Cangas do Morrazo, 36940 Cangas, Spain; 6Departamento de Bioquímica, Xenética e Inmunoloxía, Facultade de Bioloxía, Universidade de Vigo, Rúa Fonte das Abelleiras s/n, 36310 Vigo, Spain; 7Centro de Investigación Mariña, Universidade de Vigo, 36310 Vigo, Spain

**Keywords:** barracudina, coastal waters, paralepidids, feed, reproduction

## Abstract

**Simple Summary:**

The family Paralepididae is currently composed of an undetermined number of small to medium-sized, very elongated and slender aulopiform fishes distributed throughout the world, which makes it taxonomically complex. The genus *Magnisudis* is a poorly characterized group of paralepidid fishes with three recognized species. In the present study, we analyze DNA sequences deposited in public repositories to infer the taxonomic composition of the genus *Magnisudis* and its relationships with other taxonomic groups within the family Paralepididae. Morphological and molecular evidence support the identification of the specimen captured close to Galician coast (NW Spain) as *Magnisudis atlantica*. The feeding and reproductive phase of this specimen are studied and considered according to current knowledge of the species.

**Abstract:**

One specimen of the duckbill barracudina *Magnisudis atlantica* of 402 mm TL was caught in a shallow coastal area in Galician waters, northwest of Spain. Morphometric and meristic parameters along with DNA barcoding, based on cytochrome c oxidase subunit I, were used to confirm the specimen identity. Neighbor-joining analysis of nominal sequences of the genus *Magnisudis* obtained from the Barcode of Life Data System indicates the presence of six representative groupings of potential species, in contrast to the three that are currently recognized as valid. The stomach contents showed remains of digested crustaceans, tentatively identified as Euphausiids. Histological examination of the gonads revealed the specimen to be an immature female with oocytes at the primary growth stage, indicative of a lack of hermaphroditism. The results add new biological and taxonomic data that contribute to improved understanding of these poorly characterized, mainly deep-water species, demonstrating, once again, the effectiveness of DNA barcoding for identifying deep-sea fishes and characterizing their genetic differences.

## 1. Introduction

DNA barcoding has emerged as an effective tool for species identification [1], providing new insights into the study of organism biodiversity [2,3]. Its use in taxonomy, together with the traditional uses of this discipline, has allowed for a more accurate approach to taxonomic hypotheses in an integrative context [4,5]. The 5’ region of subunit I of the mitochondrial cytochrome c oxidase (*COI*) gene is the standard marker used by the scientific community for DNA barcoding. The Barcode of Life Data System (BOLD www.barcodinglife.org, accessed on 28 November 2022), is an informatics workbench aiding the acquisition, storage, analysis, and publication of DNA barcode records [6]. This portal serves as a species-level taxonomic register for the animal kingdom according to a Barcode Index Number (BIN), which is based on the analysis of patterns of nucleotide variation in the barcode region using the RESL algorithm. The latter combines single linkage clustering as a tool for the preliminary assignment of records to a presumptive species or operational taxonomic unit (OTU), employing a 2.2% value of uncorrected pairwise distance (p-distance) as the threshold, with a graph-based algorithm step called Markov clustering [7]. The Fish Barcode of Life (FISH-BOL) campaign is an international research collaboration that was launched with the aim of assembling a standardized reference DNA library for all fishes, including *COI* gene sequences from voucher specimens with authoritative identifications [8]. At that time, the results indicated that barcodes could be used to identify about 98% of already-described marine fish species. The benefits of barcoding fishes include facilitating species identification, highlighting cases of range expansion for known species, flagging previously overlooked species, and enabling identifications where traditional methods cannot be applied. Regarding fish taxonomy, one of the least studied groups is the deep-sea fish, which are species that live at depths greater than 200 m, beyond the effective range of solar radiation [9]. The increase in the pool of reference sequences of these organisms will contribute to more accurate species identification, the discovery of cryptic species, and improved understanding of intraspecific genetic differences between widely distributed species [10].

The barracudinas family Paralepididae is represented by small to medium-sized elongate fishes found throughout the world’s oceans from the Arctic to the Antarctic, although their diversity is higher in the tropics. As juveniles, these fish are epipelagic, becoming mesopelagic as adults. The taxonomic composition of the family is complex, and there is no agreement in the number of genera and species to be included, with a wide range of variation that reflects the scant knowledge currently available on this group. According to different authors, the composition could include 7 genera and 48 species [11], 7 genera and 27 species [12], 12 genera and 50 to 55 species [13], or 12 genera and 71 species [14].

The genus *Magnisudis* was established by Harry [11] when naming *Magnisudis barysoma* Harry, 1953, a junior synonym of *Magnisudis atlantica* (Krøyer, 1868). The genus at present comprises three valid species: the duckbill barracudina *M. atlantica*, the indica barracudina *Magnisudis indica* (Ege, 1953), and the Southern barracudina *Magnisudis prionosa* (Rofen, 1963). However, the validity of this genus has been questioned by Ho et al. [15], who consider the diagnostic characteristics employed to distinguish *Magnisudis* from *Paralepis* Cuvier, 1816, to be inappropriate, highlighting the need to deepen studies on the relationship between these two genera.

*Magnisudis atlantica* was originally described as *Paralepis atlanticus* on the basis of a 50.4 cm specimen that washed up at Skagen in May 1865 [16]. It is an oceanic, meso- and bathypelagic species with a depth range extending from about 50 m down to 3000 m, but it can be probably found as deep as about 5000 m [17]. It is widely distributed in tropical and temperate parts of Atlantic and Pacific Oceans but absent in the eastern tropical Pacific Ocean; specimens from the Indian Ocean, previously considered a separate subspecies, are now considered to belong to a different species. In the North Atlantic, *M. atlantica* is found from Greenland and Iceland to central equatorial waters, being rare or absent in central tropical waters and the western Caribbean Sea [18]. Only two specimens of 21 and 42 cm TL have been previously reported in Galician waters, at a depth of 764 and 892 m, respectively, in the Bank of Galicia [19].

The diet composition of *M. atlantica* consists mainly of euphausiid shrimps and pelagic fishes [20]. In turn, this species is an important prey item for large marine predators, being found in the stomach content of several thunnids, sharks, swordfish, and the long-snouted lancetfish [17,21]. Details of its reproductive mode are scarce. Some authors consider all paralepidids hermaphroditic, whereas others note sex differentiation, which could suggest varying reproductive strategies [22]; according to Russell [13], some species have separate sexes while others are synchronous hermaphrodites.

Molecular taxonomy has been successfully implemented with traditional morphological analysis in the systematic study of fishes, although it is still underdeveloped in paralepidids. DNA barcoding has been used in the molecular identification of a few species, such as *Arctozenus australis* Ho and Duhamel, 2019 [23], *Arctozenus risso* (Bonaparte, 1840), or *M. atlantica* [24], and for phylogenetic studies [25].

The genus *Magnisudis* was taxonomically revised by Post [26] based on morphological characteristics, mainly the number of vertebrae, number of fin rays, and geographical distribution. Since then, only some additional taxonomic information has been added, mainly the description of *M. indica* from the eastern Indian Ocean off Indonesia [15] and the finding of a specimen of *M. prionosa* from New Caledonia [23].

Levels of genetic variation and species status in circumglobal or cosmopolitan mesopelagic fishes remain poorly studied, and further investigations incorporating molecular genetic data are important to establish whether an apparently broadly distributed mesopelagic fish species may represent a single species, polytypic populations within a species, or multiple closely-related species [27]. The purpose of this research is to test the DNA barcoding technique in the genus *Magnisudis* and the congruence between morphological and molecular identification approaches. Moreover, the first occurrence of *M. atlantica* in a shallow coastal area of the Atlantic European waters is described, and notes on its biology are provided.

## 2. Materials and Methods

In August 2022, a specimen of *M. atlantica* was captured in Atlantic European waters in Galicia (northwest of Spain). The specimen was examined in fresh and then preserved frozen and deposited in the fish collection of the Museum Luis Iglesias de Ciencias Naturais of Santiago de Compostela Galicia, Spain, with the reference number MHN USC 25209. The main morphometric and meristic characteristics were taken following Post [26]. Standard length (SL) and head length (HL) were used throughout.

The gonad was removed and fixed in 4% neutral phosphate buffered formalin. The whole gonad was cut into four portions, two per lobule (anterior–posterior), and processed (Leica TP1020 automatic tissue processor) following the standard histological procedure of the laboratory (ethanol in increasing concentrations, Histoclear© and paraffin); sectioned at 3 μm (Microtome Leica Histocore Autocut) and stained (Leica Autostainer XL) following a routine hematoxylin–eosin staining protocol. The specimen was examined with a microscope (Leica DMRE), then sexed and classified within its correspondent maturity phase based on histological criteria [28].

The stomach was removed from the abdominal cavity and then transferred to a jar containing alcohol. Under a stereomicroscope, prey items were identified to the lowest possible taxonomic level.

The taxonomic status of the *M. atlantica* specimen captured in the waters of northwestern Spain was assessed by means of DNA barcoding. A sample of muscle was employed for DNA purification and sequencing of the standard 5’-region of the mitochondrial *COI* gene following previously described procedures [29]. PCR amplification was carried out with Thermo Scientific Phire Green Hot Start II PCR Master Mix and the primer set C_FishF1t1-C_FishR1t1 [30]. The obtained amplicon was sequenced in both directions, obtaining a sequence of 652 nucleotides in length that was registered, together with other metadata concerning the capture of the specimen and its photograph, in the BOLD database with Process ID FIGAL051-22; the sequence was also deposited in GenBank with the accession number OP575326.

The initial evaluation of the 252 public DNA barcodes deposited in the BOLD database of the family Paralepididae was carried out by aligning the 240 sequences consisting of at least 500 nucleotides in length. To check the consistency of identifications, a taxonomic cladogram was constructed using the MEGA version 11 software [31] and the neighbor-joining (NJ) method [32]. A second cladogram was constructed, taking into account only nominal sequences of the genus *Magnisudis*, including the newly obtained sequence and those grouped together in the first cladogram. The final dataset contains 41 *COI* sequences with a total of 651 nucleotide positions. The divergence among sequences was calculated using the p-distance method [33]. The percentage of replicate trees in which associated sequences clustered together was obtained by means of a bootstrap test (2000 replicates) [34].

## 3. Results

One specimen of *M. atlantica* of 402 mm TL (Figure 1a) and 195.8 g weight was caught with trammel nets on 22 August 2022, near Cíes Islands in the mouth of the Ría de Vigo (South Galicia, Northwest of Spain) at 42.2417°N, −8.9258°W and at 12 m depth. It is characterized by a moderately slender body, body depth of 7.2 in SL; long and pointed head (Figure 1b), length of 4 in SL; slender and pointed snout, 2.5 in HL; large eye with well-developed anterior adipose eyelid, bony diameter 4 in HL; large mouth extending to under the middle of the eye, with lower jaw distinctly longer than upper jaw and slightly upturned at tip; dorsal fin slightly anterior to pelvic fin insertion; well-developed dorsal adipose fin located over the base of last anal fin rays, with base slightly smaller than eye diameter; short anal fin base 7.1 in SL; depth of caudal peduncle equal to eye diameter and 27.8 in SL; pectoral fin set low on moderately long body, 8 in SL; small teeth on jaws, about 63 in the upper jaw and 40 in the lower jaw; palatines anteriorly with 4 straight canines, the third 2 mm long and the longest in the mouth, followed posteriorly by 20 small canines in a single row; no teeth on vomer; tongue with a row of eight tiny canines and one further inward on each side; gill rakers with long filaments, each raker armed with a cluster of four to seven needle-like teeth roughly arranged in two rows, those on outer row being longer (Figure 1c); and brown body in different shades.

Although all stomach contents were in an advanced stage of digestion, some observed structures allowed for identification of several characteristics corresponding to the order Euphausiacea, most probably of the family Euphausiidae (Figure 1d). The characteristic exposed specialized gills of the Euphausiacea are clearly visible even at this level of digestion. Parts of the abdominal segments and pleopods, proportionally large and rounded eyes, and heavily pigmented thoracic limbs were also visible, all consistent with identification of the prey as Euphausiacea, without any traces suggestive of identity as other taxa.

The main morphometric and meristic characteristics are shown in Table 1.

Appendix A shows an NJ tree of the paralepidid sequences available in BOLD at least 500 nucleotides in length. Most of the *Magnisudis* sequences constitute a monophyletic group at the top of this cladogram. The DNA barcodes of *Magnisudis* are well distanced from those of *Paralepis*, with the smallest genetic distance between species of these genera being 0.1377.

The NJ algorithm divides the 41 *Magnisudis* sequences into seven groups, of which six matched BOLD BINs (Figure 2). The *COI* sequence obtained from the specimen captured in the Vigo estuary, and classified by morphological features as *M. atlantica*, was grouped with 11 others within BIN AAB9276, one of whose sequences is incorrectly identified as *A. risso*, another member of the family Paralepididae. The maximum intraspecific distance in this group is 0.0184 (Table 2), well above the range of values found in the other six groups (0–0.0077). The closest group is BIN AAB9278, at a minimum interspecific distance of 0.0246, consisting of two sequences also identified as *M. atlantica.*

Both BIN AAD4540 and BIN AAD4541 groups contain sequences almost all identified as *M. prionosa* and separated by a minimum interspecific distance of 0.0380. The group formed by BIN AAD8784 consists of ten *Magnisudis* sequences identified down to genus level, including two incorrect identifications, *Paralepis elongata* (Brauer, 1906) and *A. risso*, whose maximum intraspecific distance is 0.0077 and minimum interspecific distance is 0.0523. BIN AAM0828 consists of four sequences, three of which are identified down to genus level, while the remaining one is a case of misidentification as *M. prionosa*; the distance of this group to the closest BIN is 0.0906.

Among the set of barcode sequences corresponding to the family Paralepididae found in BOLD, there are three identified as *M. atlantica* and included in the phenogram, where they cluster together and are found from the above BINs at distances ranging from 0.1525 (to BIN AAD8784) to 0.1695 (to BIN AAM0828).

An analysis of the cellular structures of the ovary allows for a basic classification of the stage of oocyte development (Figure 3). All the oocytes in the gonad were at the primary growth stage, with no evidence of secondary growth (cortical alveoli and vitellogenic oocytes), maturation (germinal vesicle migration to hydrated oocytes), or spawning events (postovulatory follicles). Thus, it was possible to determine that the examined specimen is an immature female.

## 4. Discussion

Meristic and biometric measures of this coastal record are in agreement with previous diagnoses and descriptions of *M. atlantica* [20,26].

The calculation of genetic distances between DNA barcodes stored in BOLD, and their representation in neighbor-joining clustering analysis suggests the existence of six *Magnisudis* lineages, which corresponds to a doubling in the number of known valid species of this genus. To our knowledge, this is the first study to report *COI* differentiation for *Magnisudis* using all available barcodes. The DNA barcode obtained in this research from a specimen classified as *M. atlantica* grouped with the majority of sequences that received this identification (BIN AAB9276) and should, therefore, correspond to the nominal species. Thus, the identification of record FIGAL051-22 can be confirmed using traditional and molecular taxonomic methods.

Two other sequences identified as *M. atlantica* grouped separately from the previous sequences, at a minimum distance of 0.0246 defining BIN AAB9278. In this sense, BINs AAB9276 and AAB9278 would constitute sister clades with reciprocal monophyly in the NJ cladogram, although the second grouping would contain only two individuals. The composition of both clades suggests differentiation between members of the same nominal species but in different ocean basins, since the BIN AAB9276 records are from the Atlantic Ocean while the BIN AAB9278 records are from the California coast in the Pacific. A similar previously reported observation resulted from the analysis of DNA barcodes from mesopelagic and bathypelagic teleost fishes collected in Atlantic Canadian waters [24]. Finally, three other *M. atlantica* nominal records available in BOLD (UKFBJ1219-08, ANGBF31730-19, and ANGBF31731) were grouped with species of the genus *Sudis* and are therefore considered misidentifications.

Sequences of the nominal species *M. prionosa* represented in the NJ cladogram grouped mostly in BIN AAD4540 (eight sequences), but also in BIN AAD4541 (two sequences) and BIN AAM0828 (one sequence). It seems safe to assume that only one of the first two clusters contains the sequences of individuals correctly classified as *M. prionosa*, so the others should correspond to another valid species. From the number of sequences, it is tempting to consider those present in BIN AAD4540 as the representatives of the aforementioned species.

BIN AAD8784 and BIN AAM0828 contain sequences that have, for the most part, only been identified down to genus level. The genetic distance that separates them from each other and from the rest of the clusters could grant them the rank of putative species.

Thus, only two of the six clades generated by the distance algorithm would correspond to nominal species in a genus containing three recognized species and, at the same time, with several clades containing the same nominal species. Haplotype sharing between separate “valid” species could reflect incongruence between gene trees and species (different species retaining polymorphisms from the ancestral species from which they originate but more likely due to misidentification/mislabeling of the deposited sequences [35]. However, the presence of twice as many BINs as valid species, some of which included specimens identified only down to the genus level, seems to indicate the presence of hidden biodiversity.

Ho et al. [15] questioned the distinction between *Paralepis* and *Magnisudis* based on the absence of adequate diagnostic characteristics to distinguish these two genera. Molecular data, however, confirm the validity of both genera. The DNA barcodes of *Magnisudis* are well-distanced from those of *Paralepis*, in accordance to what is expected for different genera [8].

It is clear that further accumulation of DNA barcodes from paralepidids in general and from specimens of the genus *Magnisudis* in particular is needed to determine intraspecific variation in the genus and to address, with greater certainty, the potential problems of crypticism apparently inferred by the results of this research.

Regarding biological data, *M. atlantica* is a carnivorous species, feeding particularly on young fishes and euphausiid shrimps [21,36], but for which no proper scientific feeding studies have ever been carried out [17]. However, Jones [37] confirms through feeding studies that the diet of *M. atlantica* consists of euphausiid shrimps, fish, and cephalopods, with euphausiids being the major component in the diet of adults, which is in accordance with the prey items found in the Galician specimen.

Although some authors claim that paralepidids are simultaneous hermaphrodites, a reproductive strategy often found in Aulopiform fishes [38], we found only oocytes in our sample. This suggests the occurrence of sex differentiation in this species, as has been reported for some paralepidids [39]. This is, to our knowledge, the first use of histological techniques to study the reproductive biology of *M. atlantica*, but more specimens should be examined to confirm the reproductive strategy.

Considering the oceanic character of this species, one of the most remarkable results of this finding is probably the shallow, coastal nature of the catch. Post [36] and Carl and Nielsen [17] reported that some adult individuals were approaching the coast. Excluding specimens found ashore, the 12 m depth of the Galician specimen is, as far as we know, the shallowest coastal depth reported for an adult specimen. Other oceanic epipelagic to bathypelagic fishes, such as *Schedophilus medusophagus* (Cocco, 1839) or *Alepisaurus ferox* Lowe, 1833, have also been sporadically found in coastal Galician waters [29,40].

The lack of gill rakers and teeth has turned out to be a common feature of many alepisauroid species, indicating maturity of the specimen, although this has not been verified in adults of *Magnisudis* species [26]. However, Ho and Duhamel [23] reported the reduction of teeth and gill rakers in one adult of *M. prionosa* of ca. 562 mm SL as a character of sexual maturity, suggesting that the loss of gill rakers and/or teeth in adults is common in the subfamily Paralepidinae, although it does not occur in all members [15]. In the examined *M. atlantica* specimen, no loss of gill rakers or teeth was observed, but it is considered immature despite its large size. The maturity stage of the aforementioned *M. prionosa* specimen was not provided, but given its size is the maximum reported for this species, it seems safe to assume that it was a mature specimen. The maximum known size of *M. atlantica* is 560 mm SL [20], and given that the 361 mm SL specimen is considered immature, a late first maturity size is probable. Therefore, if the lack of gill rakers and teeth are related to maturity in alepisauroid species, and this is attained at larger and less frequent sizes in individuals of *Magnisudis* species, it might better explain why this loss has hardly been observed in specimens of this genus. If this hypothesis proves to be true, the non-loss of gill rakers and teeth in adults of *Magnisudis* would have to be discarded as a taxonomic characteristic, as proposed in Post [26].

## 5. Conclusions

DNA barcoding was used to validate the genus *Magnisudis*, which is clearly differentiated from *Paralepis*, but the number of clades resulting in the neighbor-joining analysis of the former is twice the number of recognized species, suggesting hidden biodiversity. Morphological analysis of a specimen of *Magnisudis atlantica* caught in Atlantic Spanish coastal waters confirms the unusual presence of the species in this habitat. The remains of Euphausiidae prey found in the stomach contents are consistent with its consideration as the most common prey in the diet. In contrast, reproductive histological analysis showed sexual differentiation instead of the simultaneous hermaphroditism reported to date.

## Figures and Tables

**Figure 1 biology-12-00349-f001:**
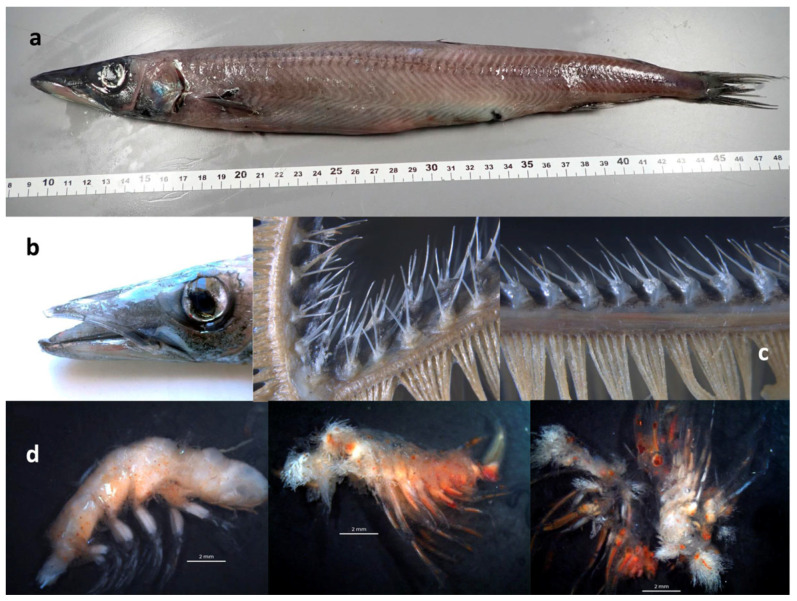
(**a**) *Magnisudis atlantica* MHN USC 25209, 402 mm TL. (**b**) Detail of mouth and head. (**c**) Detail of the gill rakers showing needle-like teeth. (**d**) Remains of Euphausiids prey items.

**Figure 2 biology-12-00349-f002:**
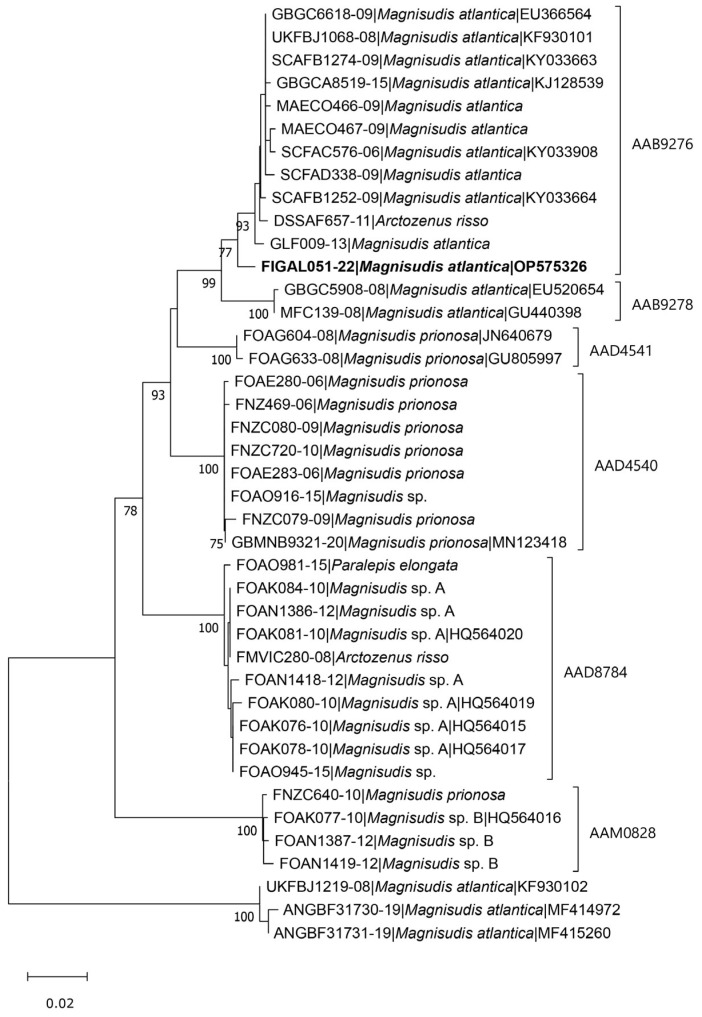
Neighbor-joining representation of *Magnisudis COI* sequences. The percentage of replica trees in which associated sequences cluster together in a bootstrap test (2000 replicates), as shown below the branches. The bar shows the unit for the number of base differences per site (p-distance). The analysis included 41 nucleotide sequences. All ambiguous positions were removed for each sequence pair (pairwise deletion option). There were a total of 651 positions in the final dataset. The Spanish coastal record is represented in bold type. The diagram includes the correspondence between each grouping and its BOLD BIN.

**Figure 3 biology-12-00349-f003:**
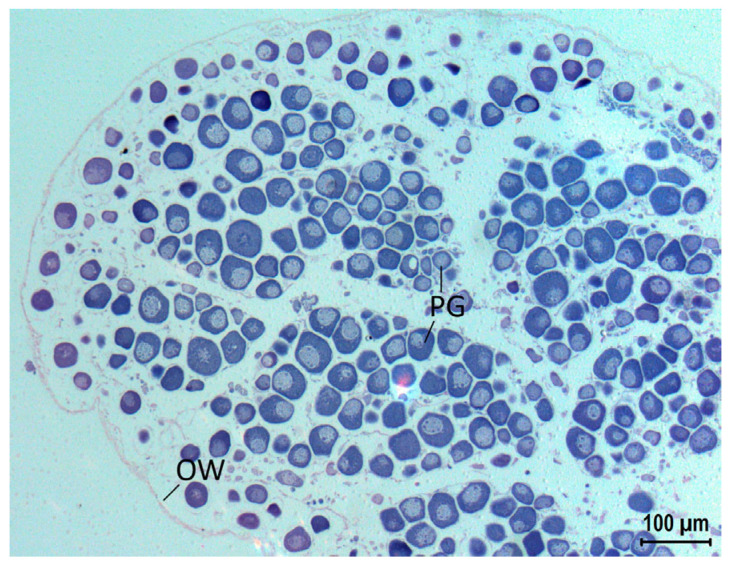
Histological slide of the ovary of *Magnisudis atlantica* MHN USC 25209, 402 mm TL (magnification 10×); PG = primary growth oocyte; OW = ovarian wall.

**Table 1 biology-12-00349-t001:** Morphometric and meristic data of *Magnisudis atlantica* MHN USC 25209.

	L (mm)	%SL	%HL	in SL	in HL
Total length	402				
Standard length	361				
Pre-dorsal fin length	212	58.7		1.7	
Pre-pelvic fin length	220	60.9		1.6	
Pre-anal fin length	280	77.6		1.3	
Pre-adipose fin length	314	87.0		1.1	
Interdorsal adipose length	81	22.4		4.5	
Pre-anus length	232	64.3		1.6	
Head length	89	24.7		4.1	
Head depth	37	10.2		9.8	
Snout length	35	9.7	39.3	10.3	2.5
Post-orbital length	35	9.7	39.3	10.3	2.5
Pre-nostril length	24	6.6	27	15	3.7
Upper jaw length	38	10.5	42.7	9.5	2.3
Lower jaw length	40	11.1	44.9	9.0	2.2
Eye diameter	22	6.1	24.7	16.4	4.0
Interorbital width	14	3.9	15.7	25.8	6.4
Mouth gape	40	11.1	44.9	9.0	2.2
Body depth at pectoral fin origin	42	11.6	47.2	8.6	2.1
Body depth (maximum)	50	13.9	56.2	7.2	1.8
Anal fin base	51	14.1	57.3	7.1	1.7
Dorsal fin base	19	5.3	21.3	19.0	4.7
Adipose fin base	9	2.5	10.1	40.1	9.9
Pectoral fin length	45	12.5	50.6	8.0	2.0
Pelvic fin length	14	3.9	15.7	25.8	6.4
Caudal peduncle length	13	3.6	14.6	27.8	6.8
Caudal peduncle depth	13	3.6	14.6	27.8	6.8
Meristic					
Dorsal fin rays	10				
Anal fin rays	23				
Pectoral fin rays	17				
Pelvic fin rays	9				
Branchiostegal rays	6				
Gill rakers	7 + 29				
Premaxillary teeth	63				
Dentary teeth	40				
Lateral line scales	63				

**Table 2 biology-12-00349-t002:** Maximum intraspecific and minimum interspecific p-distances in *Magnisudis* groups.

BIN	N	Maximum Intraspecific	Minimum Interspecific					
			AAB9276	AAB9278	AAD4541	AAD4540	AAD8784	AAM0828
AAB9276	12	0.0184						
AAB9278	2	0.000	0.0246					
AAD4541	2	0.0017	0.0459	0.0522				
AAD4540	8	0.0048	0.0476	0.0492	0.0380			
AAD8784	10	0.0077	0.0691	0.0737	0.0617	0.0523		
AAM0828	4	0.0046	0.0983	0.1045	0.0934	0.0906	0.0906	
Unknown	3	0.0016	0.1649	0.1690	0.1614	0.1633	0.1525	0.1695

## Data Availability

The sequences employed in the current study are available in the BOLD systems (https://www.boldsystems.org/, accessed on 1 December 2022) and GenBank (https://www.ncbi.nlm.nih.gov/genbank/, accessed on 1 December 2022) repositories. All specimens used in this study for taxonomical purposes are deposited in the fish collection of the Museo de Historia Natural, Universidade de Santiago de Compostela (MHNUSC) in Santiago de Compostela, Spain (see methods). All other data are included in this article.

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
