# Peer review of "DNA Barcoding of the Genus Magnisudis (Aulopiformes: Paralepididae) with a Coastal Record and Biological Features of Magnisudis atlantica"

_biology, 2023, doi:10.3390/biology12030349_

Round 1

Reviewer 1 Report

This manuscript describes the opportunistic discovery of a duckbill barracudina, a species that belongs to a little-studied group of fishes that may contain cryptic diversity. To confirm specimen identity, the authors use both DNA barcoding and morphological characteristics. They also present some natural history data regarding diet and reproduction in the poorly-known species. Finally, they present some results for DNA barcodes and suggest there may be cryptic species diversity in the group.

I find this paper to be a great example of multiple methods to confirm identity of a cryptic species and to use an opportunistic finding to gather as much natural history data as possible. I do have two primary question/concerns with the manuscript. First, I think a mitochondrial locus is not an appropriate basis for delineating species. To be fair, the authors do not attempt to designate new species and the results are suggestive of cryptic diversity, but I think it would be important for the authors to note that additional loci and sequencing would be needed to test the hypothesis of six species within this group. But more broadly, given that all of these sequences except one are already known, and the introduction indicates that the systematics of this group has been investigated (108-111), I would like to see more detail in the introduction about how species have been described. Was it primarily based on morphology? Were most of these sequences collected after taxonomic investigation? Has COI differentiation already been noted in the literature? This would really help with the context of why the authors did the neighbor-joining tree and why potential cryptic diversity hasn’t already been identified.

My second comment is related to the diet identification. Given that work was ongoing to sequence the fish species and the partially digested remains of the potential Euphaussid prey, why was sequencing not performed to identify this prey species? Are there not sequences available for species in this group? While I recognize it is not probably not realistic for this analysis to occur at this point, perhaps the authors could note that barcoding could also be used for greater confidence in dietary analysis.

Author Response

I do have two primary question/concerns with the manuscript. First, I think a mitochondrial locus is not an appropriate basis for delineating species. To be fair, the authors do not attempt to designate new species and the results are suggestive of cryptic diversity, but I think it would be important for the authors to note that additional loci and sequencing would be needed to test the hypothesis of six species within this group.

Yes, we know that the mitochondrial locus is not an appropriate basis for species delimitation, but as the reviewer says, this is not a species-delimitation manuscript. This was not the aim but rather a derived consequence of this research. However, we do provide new information that can serve as a basis for undertaking more in-depth analyses in the future. Although DNA barcoding represents a powerful triage tool for biodiversity assessment, and short-length single-locus markers as COI are often effective proxies for species, they are frequently not representative of full phylogenetic history. Of course, subsequent to an initial DNA barcode triage, COI data can then be incorporated into more sophisticated species delimitation systems using multiple loci (all this reviewed and summarised in Collins, R.A. and Cruickshank, R.H. (2012) The seven deadly sins of DNA barcoding. Molecular Ecology Resources 13, 969-975).

But more broadly, given that all of these sequences except one are already known, and the introduction indicates that the systematics of this group has been investigated (108-111), I would like to see more detail in the introduction about how species have been described. Was it primarily based on morphology?

Yes, the original taxonomic descriptions of fishes and the subsequent taxonomic revisions are almost all based on morphological characters, as molecular taxonomy is very recent. We have included a clarification regarding the main morphological characters in Magnisudis

Were most of these sequences collected after taxonomic investigation?

Our data is primarily based on a morphological identification. We use our own information to compare with the previous available data from public databases, which has various origins, not all of them morphologically contrasted. This has allowed us to detect errors in the identification of sequences, as well as to confirm others, but it is not always possible to know to what extent each of the previous samples has been studied at a morphological level and at what level of detail.

Has COI differentiation already been noted in the literature?

To our knowledge, this is the first study to report COI differentiation for Magnisudis species, at least with all available barcodes, but this technique is frequently used for this purpose in fishes. We have included a small explanatory paragraph to give a little more background information on this matter.

My second comment is related to the diet identification. Given that work was ongoing to sequence the fish species and the partially digested remains of the potential Euphaussid prey, why was sequencing not performed to identify this prey species? Are there not sequences available for species in this group? While I recognize it is not probably not realistic for this analysis to occur at this point, perhaps the authors could note that barcoding could also be used for greater confidence in dietary analysis.

We agree that it would be interesting to sequence the stomach remains and compare them with the available information. However, we consider that identify the remains to species level is not of primary relevance. We only expect to get an idea of the type of prey they are looking for, based on what is left in the stomach. Even if we see euphasiids, it is not expected that the prey species is always the same. It varies between different areas, so what we consider relevant is the type of organism, which gives an idea of the biological aspects that characterize the predated species (size, shape, swimming capability or nutritional value).

Reviewer 2 Report

After reading in detail the manuscript, I have no commnet on it. It is very well written, and it has a high quality in both methodology and results obtained. I have no hesistation to accept the manuscript as it has been submitted. The Fishures are good quality and reflect the results got by the authors.

Author Response

After reading in detail the manuscript, I have no comment on it. It is very well written, and it has a high quality in both methodology and results obtained. I have no hesitation to accept the manuscript as it has been submitted. The Figures are good quality and reflect the results got by the authors.

Thank you very much